# Human Joint Torque Estimation Based on Mechanomyography for Upper Extremity Exosuit

Yongjun Shi [1], Wei Dong [1], Weiqi Lin [1], Long He [2], Xinrui Wang [2], Pengjie Li [2] and Yongzhuo Gao [1,*]

[1] State Key Laboratory of Robotics and System, Harbin Institute of Technology (HIT), Harbin 150001, China; 19b908038@stu.hit.edu.cn (Y.S.); dongwei@hit.edu.cn (W.D.); 22b308009@stu.hit.edu.cn (W.L.)
[2] Weapon Equipment Research Institute, China South Industries Group Corporation, Beijing 102202, China; helong208@126.com (L.H.); xinruii@163.com (X.W.); lpj208@163.com (P.L.)
* Correspondence: gaoyongzhuo@hit.edu.cn

**Abstract:** Human intention recognition belongs to the algorithm basis for exoskeleton robots to generate synergic movements and provide corresponding assistance. In this article, we acquire and analyze the mechanomyography (MMG) to estimate the current joint torque and apply this method to the rehabilitation training research of the upper extremity exosuit. In order to obtain relatively pure biological signals, a MMG processing method based on the Hilbert-Huang Transform (HHT) is proposed to eliminate the mixed noise and motion artifacts. After extracting features and forming the dataset, a random forest regression (RFR) model is designed to build the mapping relationship between MMG and human joint output through offline learning. In addition, an upper extremity exosuit is constructed for multi-joint assistance. Based on the above research, we develop a torque estimation-based control strategy and make it responsible for the intention understanding and motion servo of this customized system. Finally, an actual test verifies the accuracy and reliability of this recognition algorithm, and an efficiency evaluation experiment also proves the feasibility for power assistance.

**Keywords:** joint torque estimation; upper extremity exosuit; mechanomyography signal processing; rehabilitation training; MMG; human movement assistance

## 1. Introduction

As is the case for most of the potential technical equipment for rehabilitation training and movement assistance, the exoskeleton system has always attracted attention in related research fields for the elderly and the disabled [1,2]; however, with some key issues not being resolved, the performance of this wearable robot remains relatively limited [3]. Among them, human intention perception belongs to a pretty critical research point and needs to be studied further. Traditional methods try to achieve this function by monitoring limb kinematics data [4] or human-machine interaction information [5], but problems such as response lag greatly restrict its actual effect. In recent years, recognition methods based on biological signals have emerged and gained wide attention [6], which are expected to realize more effective intention understanding.

Biological signals are often generated before the execution of corresponding actions, which show a certain degree of motion predictability and can make intention recognition more timely and accurate. In addition to eye tracking and galvanic skin response (GSR) [7], which are not suitable for combining with exoskeleton control technology, commonly used types in current research mainly contain electroencephalogram (EEG) [8], surface electromyogram (sEMG) [9], and mechanomyography [10]. The EEG signal originates from the potential of the external electrical field that fluctuates around nerve cells in the brain, and is often applied to classify specific movement patterns [11,12]. Due to its instability and susceptibility to interference, EEG-based methods for data collection, signal processing, and intent identification needs to be further improved and optimized. The sEMG signal

represents the total action potentials of different motor units innervated by certain motor neurons, which can be obtained through electrodes placed on the corresponding muscle tissue [13]. It has the advantage of high sensitivity and low delay, but many factors will diminish its effectiveness for data collection, such as skin surface cleanliness, air humidity, and patch electrode position [14]. MMG reflects the mechanical vibration signal generated when the skeletal muscle contracts, and it contains abundant information, such as the number of muscle fibers participating in the exercise, and the amplitude and frequency of vibration [15]. Compared with the above two signals, it has the following advantages [16].

- The measuring device can be used directly without touching the skin, which simplifies the preparation before data collection;
- the signal has strong anti-interference ability and will not be affected by environmental variables such as sweat, humidity, or electromagnetics;
- the equipment cost is pretty low because the data collection task can be completed by using an acceleration sensor that meets the accuracy requirements.

However, there are still some limitations in the practical application of MMG. For example, it is easily contaminated by low-frequency motion artifacts, and sensitive to sudden step noises; therefore, a suitable signal processing algorithm is needed to extract a relatively pure sequence.

At present, classification and regression algorithms have been widely used in many studies, such as the trajectory control of a redundant robot [17] and hand gesture recognition for teleoperated surgical robot systems [18], which can also complete the analysis of human motion intention. As a relatively common and easy-to-use method, motion pattern recognition [19,20] aims at establishing the mapping relationship from sensor information to finite human motion states through classification models. Then, the control subsystem will generate corresponding commands according to current motion pattern and will deliver them to underlying drivers. Joint angle prediction attempts to calculate the limb position at the next moment based on regression models [21,22], which can effectively avoid the response lag of the exosuit. After that, the motion state of the power-assisted system can be dynamically adjusted through position closed-loop control. Joint torque estimation also belongs to a direct and effective method of obtaining intentions [23,24]. The desired torque can be calculated through the Hill type model [25] or obtained from the biological signals using machine learning algorithms [26], and can act as the input parameter of torque closed-loop control to regulate the motor output. The last method will provide a reference for the direct control of external torque that assists joint motion, which is quite practical for upper extremity exosuits that need to achieve an expected power-assisted efficiency.

If we want to use the abovementioned MMG signal to estimate the joint torque, it is obviously quite difficult through conventional mathematical derivation. The machine learning algorithm can train the mapping model very well based on the existing data, and it shows an excellent fitting ability in many research fields, such as breathing pattern detection [27] and human activity identification [28]; therefore, this method should be able to describe the complex and nonlinear relationship between MMG and joint torques.

The inherent characteristics of the soft exosuit based on Bowden cables greatly increases the difficulty for designing control strategies [29,30]. The gravity compensation algorithm is a simple and common method for motion generation, which will output an active torque to offset the joint load imposed by the limb weight [31,32]; however, due to the large error of compensation model and the ignorance of dynamic characteristics, it may cause the system response to, more or less, mismatch the joint movement. The exosuit named CRUX [33] tries to control the target arm to follow the reference trajectory of the healthy one, consequently completing the active rehabilitation training process [34]. This method will limit the subjective initiative of the wearer to a certain extent, and is not suitable for situations where both arms need assistance. Some scholars from Italy have proposed a threshold method based on sEMG [35]. When the signal amplitude of the wearer's measured muscle exceeds the set value, the wearable system starts to pro-

duce a power-assisted effect for motion immediately. The most obvious disadvantage of this method is that it only outputs a constant driving force, but the required joint torque changes dynamically under different motion states. In general, the control logic for the upper extremity exosuit still has some defects, and remains to be further explored.

In this paper, we intend to complete data preprocessing and feature extraction using MMG, then establish the mapping relationship from this signal to the joint torque based on the machine learning algorithm, and finally apply it to the rehabilitation training research of upper extremity exosuit. MMG is collected through an inertial measurement unit (IMU) and synthesized by the linear accelerations along three axes. The HHT will filter this original signal to obtain a relatively pure result. We extract three features from the processed data and combine them with the collected joint torque to form data sets for training and testing. A RFR model is designed as the algorithm framework for joint torque estimation, and its parameters are determined through offline training. According to the above research results, a control algorithm based on joint torque estimation will take charge of the motion control for an upper extremity exosuit. Eventually, some experiments are carried out to test the accuracy of torque estimation and the efficiency of power assistance. The main contributions and highlights of this study are summarized as follows.

- We have attempted to use the MMG signal as the medium for the exosuit to understand human intentions, and to demonstrate the feasibility and effectiveness of this approach.
- A series of MMG-related methods for signal acquisition, filtering, and feature extraction have been developed.
- A regression model reflecting the nonlinear relationship between muscle activation and joint output is constructed.
- A torque estimation-based control algorithm is designed and applied to the multi-joint motion assistance of upper extremity exosuit, which can significantly amplify the limb strength.

The remaining research content of this article is organized as follows. Section 2 describes the measurement and calculation methods for MMG, its corresponding joint torque, and how to process the original signal to construct data sets. Section 3 introduces the design details of RFR model and uses a large amount of test data to fit the desired quantitative relationship. In Section 4, we have developed a control strategy for the upper extremity exosuit based on torque estimation. Section 5 proves the feasibility of the above methods through some experiments. Section 6 is the conclusion.

## 2. Data Sets Acquisition

### 2.1. Raw Information Collection

In order to obtain the MMG signal and corresponding joint torque at the same time, we built a measurement platform for joint information collection. Figure 1a shows how to use this device to get relevant data about elbow static flexion and extension.

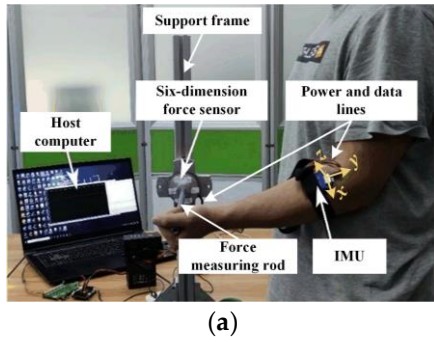
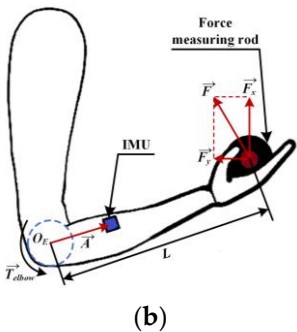

(**a**)　　　　　　　　　　　　　(**b**)

**Figure 1.** Information collection process during elbow static flexion/extension: (**a**) measurement platform; (**b**) joint torque calculation model for elbow.

During this process, a six-dimension force sensor and a high-performance IMU are responsible for detecting forces along three directions at the end of limb, and gathering MMG at the brachioradialis of forearm, respectively. When subjects hold the force measuring rod and try to perform specific actions, this platform will transmit corresponding data to the host computer through a data line and save them in files.

With readings from a six-dimension force sensor ($F_x$ and $F_y$) as input, the joint torque ($\vec{T}_{elbow}$) can be calculated by a certain mathematical relation. The mechanical model of the elbow is shown in Figure 1b, where $\vec{A}$, $\vec{F}$, and $L$, respectively, represent the current posture vector of human arm, the resultant force vector at the end, and the length of forearm. Then, we can dynamically obtain the joint torque values during elbow static flexion and extension through the following formulas.

$$\vec{F} = \vec{F}_x + \vec{F}_y \tag{1}$$

$$\vec{T}_{elbow} = L\left(\frac{\vec{A}}{|\vec{A}|} \times \vec{F}\right) \tag{2}$$

During dynamic flexion/extension without using the platform, it can be seen in Figure 2 that when the measured muscle contracts or relaxes and drives the human joint to rotate, the IMU can perceive some regular acceleration signals in the $x$, $y$, and $z$ directions synchronously, and the same is true for the process of static joint output. In order to integrate all the effective information, we calculate the sum of linear acceleration vectors along three axes ($ACC_x$, $ACC_y$, and $ACC_z$), and take its modulus as the original MMG signal ($MMG(t)$), which can be expressed as the following equation.

$$MMG(t) = \sqrt{ACC_x^2 + ACC_y^2 + ACC_z^2} \tag{3}$$

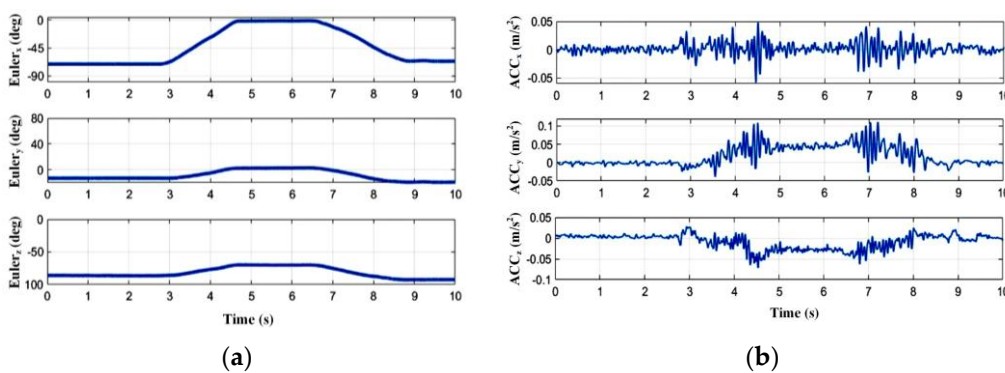

(a)  (b)

**Figure 2.** The IMU data during elbow dynamic flexion/extension with low strength: (**a**) three-axis euler angles; (**b**) three-axis accelerations.

### 2.2. MMG Signal Processing

However, the method of using linear acceleration values for MMG characterization inevitably mixes low-frequency artifacts into the collected original signal, such as gravitational acceleration, IMU posture changing artifacts caused by muscle deformation, and motion artifacts introduced by upper limb movements. Moreover, high-frequency white noise may also be superimposed on it. In order to improve the quality of data and lay the foundation for subsequent feature extraction, an effective filtering method must be applied to eliminate the abovementioned interferences. Traditional signal processing methods are mostly based on Fourier analysis, but these ones have limited effects in practical applications of processing MMG due to its non-linear and non-stationary characteristics. With reference to related literatures, we decide to use HHT to analyze the original data, which is more suitable for these kinds of signals.

The HHT consists of two parts, namely empirical mode decomposition (EMD) and the Hilbert transform. EMD can decompose a complex signal into a limited number of intrinsic mode functions (IMFs) and a residual based on the local time scale characteristics of itself. The specific implementation steps are as follows. First, we find all the local maximum points and local minimum points of the original MMG signal and fit their respective envelopes through the cubic spline curve. Then, the mean value of the upper and lower envelopes ($U(t)$ and $L(t)$) will be subtracted from the original sequence to get the remaining part with the low frequency information removed ($x(t)$).

$$x(t) = MMG(t) - \frac{1}{2}(U(t) + L(t)) \tag{4}$$

If the number of extreme values and zero crossings on the entire data set of $x(t)$ differs by 0 or 1, and the average value of its two envelopes remains zero at any time, then $IMF_i(t) = x(t)$, otherwise it is necessary to let $MMG(t) = x(t)$, and repeat the above steps until these two conditions are met. Next, we remove the obtained $IMF_i(t)$ from $MMG(t)$ and repeat all the above steps again with the remaining part ($r_i(t)$) to get other IMF components until $r_i(t)$ is a constant or monotonic function. As shown in Figure 3a, the original MMG signal is decomposed into 9 IMFs and 1 residual ($res(t)$) according to the signal frequency, which can be expressed as follows.

$$MMG(t) = \sum_{i=1}^{9} IMF_i(t) + res(t) \tag{5}$$

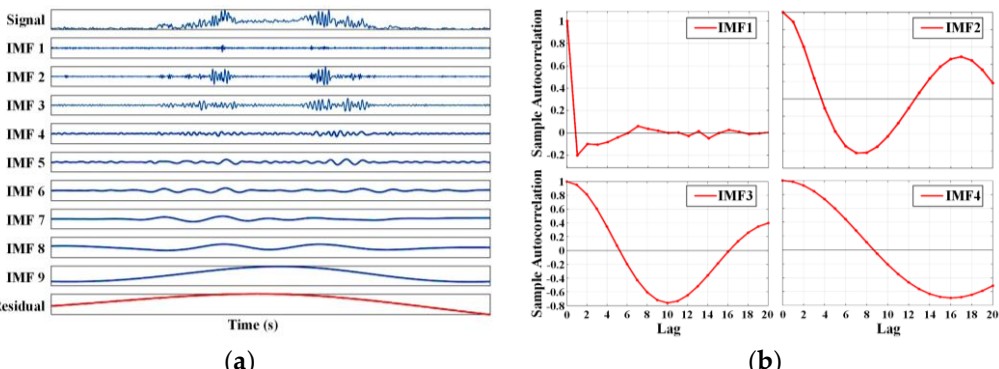

(a)  (b)

**Figure 3.** MMG signal processing: (**a**) decomposing result through EMD; (**b**) autocorrelation function curves of the first four IMFs.

After completing the above analysis, it is time to select the IMFs dominated by MMG through certain methods and reorganize them to obtain a relatively pure signal. To eliminate the noise-dominated IMFs, we introduce the concept of autocorrelation ($R_{IMF}(t_1, t_2)$), which reflects the correlation degree of signal values at different times ($t_1$ and $t_2$). Its normalized expression form, ($\rho_{IMF}(\tau)$), can be obtained with the following formula, where $\tau = t_1 - t_2$.

$$R_{IMF}(t_1, t_2) = E[IMF(t_1) \cdot IMF(t_2)] \tag{6}$$

$$\rho_{IMF}(\tau) = \frac{R_{IMF}(\tau)}{R_x(0)} \tag{7}$$

If the normalized autocorrelation curve belongs to an impulse function close to the zero point, it can be ascertained that the corresponding IMF is dominated by noise, because noise has randomness and a weak correlation at every moment. After the calculations in Figure 3b, the first IMF can be classified as such a disturbance.

To exclude the IMFs dominated by motion artifacts, we try to find the discrimination basis from the energy distribution of each order component. After calculating the energy value ($E_{IMF}^i$) of each IMF according to the following Formula (8), it is revealed that energy

contained in each IMF has increased significantly, starting from the sixth order. By analyzing the data of different participants, we find that removing these IMFs as motion artifacts can achieve a better result.

In the end, we believe that IMF2~IMF5 can effectively characterize the muscle activity, and the filtered signal can be obtained with them being recombined. It can be seen from Figure 4 that there is a pretty clear correspondence between the processed MMG and joint torque.

$$E_{IMF}^i = \frac{1}{n} \sum_{j=1}^{n} [IMF_i(j)]^2 \tag{8}$$

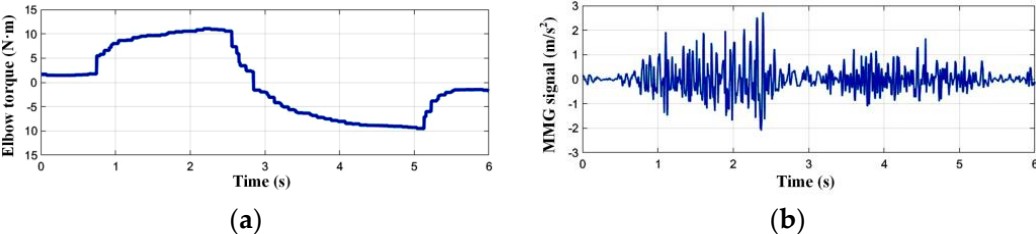

(a)                                            (b)

**Figure 4.** Comparison between the filtered MMG of the brachioradialis muscle and the corresponding elbow joint torque: (**a**) the change curve of joint torque; (**b**) the change curve of filtered MMG.

*2.3. Feature Extraction*

Considering that the change in limb strength is often accompanied by the fluctuation of the muscle fiber's vibration amplitude, we select the root mean square (RMS) as the time domain characteristic of MMG. It reflects the effective value of data amplitude and can be calculated with the following formula.

$$RMS_{MMG} = \sqrt{\frac{1}{N} \sum_{i=1}^{N} X_i^2} \tag{9}$$

Due to the correlation between muscle activity and its vibration frequency, mean power frequency (MPF) is used to represent the frequency domain characteristic of MMG. It is necessary to perform the Hilbert transform on the filtered MMG to analyze its frequency spectrum, and then integrate it on the time axis to obtain the Hilbert marginal spectrum that characterizes the relationship between signal frequency ($f_i$) and energy ($E_i$). Then, MPF can be calculated through the following equation.

$$MPF_{MMG} = \frac{\sum_{i=1}^{N} f_i E_i}{\sum_{i=1}^{N} E_i} \tag{10}$$

The MMG signal may contain information about the number of muscle fibers involved in power output; therefore, we additionally introduce the concept of sample entropy (SampEn) to describe the characteristic from a non-linear perspective, which can quantify the complexity of the time series.

In order to not lose continuity information in the sequence, we apply the sliding window strategy to extract these characteristic values of the filtered MMG signal. The window length and step length are set as 500 ms and 50 ms, respectively. So far, the data set of elbow static flexion/extension is established with RMS, MPF, and SampEn of the MMG signal as features, and joint torque as the label. We can also use similar methods to obtain relevant information of shoulder static flexion/extension and static adduction/abduction.

### 3. Off-Line Torque Estimation
*3.1. Regression Model Design*

There is a relatively complicated non-linear relationship between MMG and human joint torque, which seems difficult to accurately describe using the traditional polynomial

regression model. We try to introduce a machine learning method to solve this problem, using a large amount of test data to fit the real mapping law. Considering that the joint torque estimation algorithm is oriented to a wearable power-assisted system, the requirement for its stability and reliability must take precedence over that of other aspects. Since RFR (shown as Figure 5) has these advantages, it can be well qualified for this task.

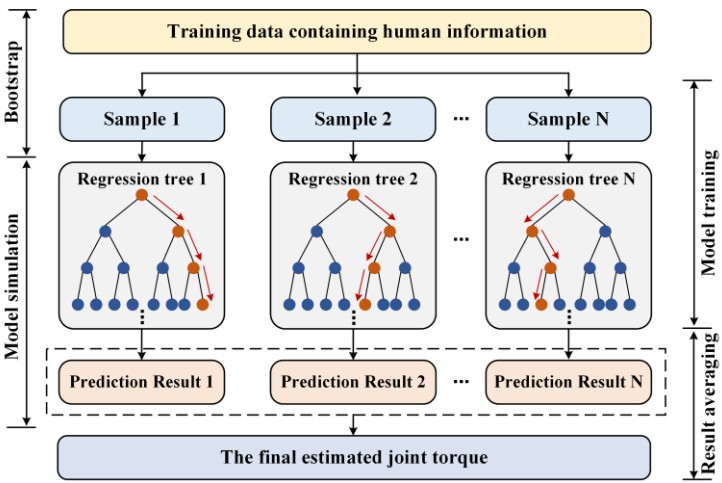

**Figure 5.** Schematic diagram of RFR algorithm.

The RFR belongs to a bagging type algorithm of ensemble learning, which aims at improving overall performance by packaging and combining several weak models, namely decision trees, into a strong one. The entire model consists of multiple classification and regression trees (CARTs) that are not related to each other. All CARTs jointly determine the final output result. The specific implementation steps of the algorithm are described as follows.

1.  Randomly extract any number of samples from the training set to form multiple new sub-training sets;
2.  use each sub-training set to train a CART separately. During this process, it is necessary to randomly obtain any number of features from all the features, and then select the optimal segmentation point to cut the subtree;
3.  repeat step 2 to obtain multiple trained CARTs;
4.  calculate the average of all the CARTs' prediction results and use it as the final estimated value.

Only when more than half of the CARTs make wrong predictions will the output of the RFR model seriously deviate from the true value. Even if an abnormal data point appears, it does not affect the performance of entire algorithm too much, which fully reflects the strong robustness to stop interference signals.

### 3.2. Off-Line Training and Testing

We recruited three healthy adult men to participate in training data acquisition. Based on the abovementioned platform and methods, the information collection experiment for three motion modes obtains 150,000 sets of sample data in total. In accordance with the idea of cross-validation, one-fifteenth of them are selected as the test set, and the remaining data act as the training set. Finally, on the basis of setting the number of sub-regression trees of the RFR to 10, and the minimum leaf size to 1, the training process has been carried out, and the verification result of the test set is also obtained.

Taking shoulder static adduction/abduction as an example, Figure 6a shows that the minimum mean square error (MSE) decreases and tends to be stable with the increase in iterations, and Figure 6b indicates that the difference between the predicted results and

the actual values on the test set is relatively small. In general, the model training effect has reached the desired level.

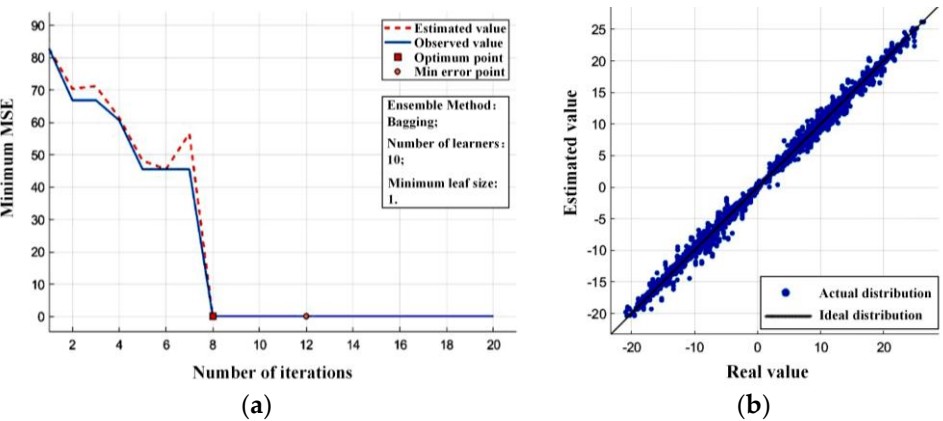

(a)                                                   (b)

**Figure 6.** Machine learning effect of shoulder static adduction/abduction: (**a**) iterative training process with training set; (**b**) verification result of test set.

To measure the predictive performance of trained RFR model, a root mean square error (*RMSE*) and a coefficient of determination ($R^2$) are introduced as evaluation indexes. The *RMSE* is a commonly used method to express numerical errors, representing the sample standard deviation of the difference between the predicted value and the actual one. It can be calculated by using the following formula.

$$RMSE = \sqrt{\frac{1}{n}\sum_{t=1}^{n}(\hat{y}_t - y_t)^2} \tag{11}$$

The $R^2$ reflects how much the regression relationship can account for changes to the dependent variable. A higher value indicates that the regression model can produce better prediction results. The corresponding calculation process is shown below.

$$R^2 = 1 - \frac{\sum_{t=1}^{n}\hat{y}_t - y_t)^2}{\sum_{t=1}^{n}(y_t - \overline{y})^2} \tag{12}$$

## 4. Test Platform Construction

### 4.1. Overview of the Upper Extremity Exosuit

We intend to take advantage of the abovementioned research to design a control logic for an upper extremity exosuit, so that it can perform rehabilitation training functions according to human intentions. As shown in Figure 7, this wearable system aims at providing active assistance for shoulder flexion/extension, shoulder adduction/abduction, and elbow flexion/extension of the left arm.

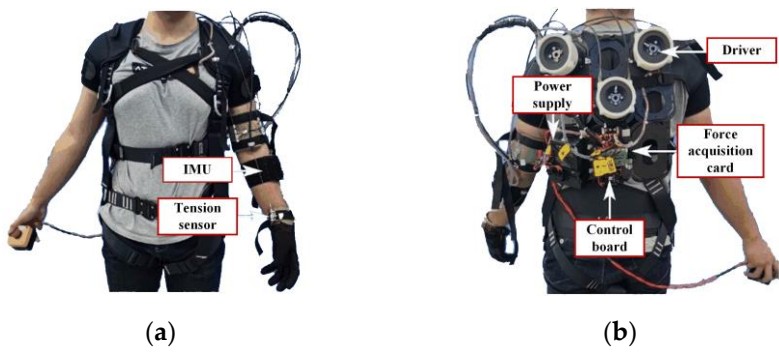

(a)                                                   (b)

**Figure 7.** Pictures of an upper extremity exosuit: (**a**) the front; (**b**) the back.

It contains three sets of cable-driven modules, each of which is responsible for driving the bidirectional motion for one degree of freedom. The sensing network consists of three IMUs, six tension sensors, and three absolute encoders which are integrated in motors, and are in charge of completing multiple tasks, such as MMG signal collection, limb posture perception, human-machine interaction information acquisition, and servo motor state reading. As the main control board, STM32F407IGH6 will serve as the brain of the system to perform core functions such as feature extraction, motion intent identification, and motor servo control. Components communicate with each other through CAN bus for data feedback and instruction delivery.

On the basis of the abovementioned hardware, the exosuit can be driven to assist the human limb coupled with a suitable control algorithm.

### 4.2. Torque Estimation-Based Control Strategy

As shown in Figure 8, the control logic framework of the upper extremity exosuit consists of two layers, namely, the intent analysis part based on torque estimation, and the motion control part based on torque closed-loop.

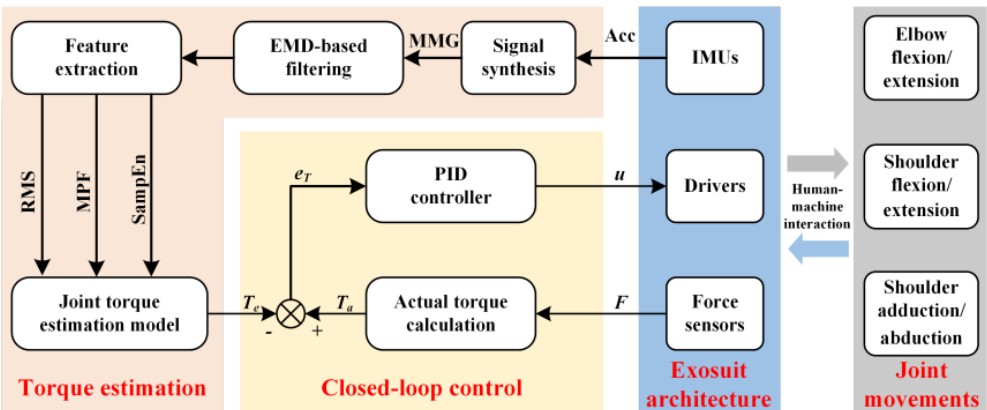

**Figure 8.** Torque estimation-based control strategy.

When subjects wear this exosuit for collaborative movement, the upper controller will obtain the triaxial accelerations from the target muscle through the accelerometer embedded in the IMU, and synthesize them into an original MMG. After completing the EMD-based filtering operation, it screens out relatively pure signals and extracts the three characteristics including RMS, MPF, and SampEn. The trained RFR model uses these features as an input to estimate the expected joint torque value at the current moment. Finally, the corresponding control commands will be sent to the lower layer.

The lower controller calculates the actual joint torque using the tension sensor readings at the end of the Bowden cable and compares it with the expected joint torque received from the upper layer to obtain their error value. Then, a standard PID algorithm generates motor drive commands based on the torque error, and controls the cable-driven module to provide appropriate assistance at the joint.

To realize this control strategy through a program code, we develop an embedded software based on the µC/OS III operating system. Five sub-tasks, including sensing data reception, signal processing, feature extraction, torque estimation, and motor servo control, are set up in order of priority from high to low. The execution frequency of each is assigned by setting different cycle times. Through the division of the abovementioned sub-task modules, we strengthen the real-time performance of programs under the premise of clarifying the control code logic for the upper extremity exosuit. In addition, it is convenient for subsequent optimization work.

## 5. Experiment on Exosuit

### 5.1. Reliability Analysis Experiment for Torque Estimation

The parameters of the RFR model are determined by offline training on a PC. After transplantation to the control system of the exosuit, an evaluation experiment needs to be carried out to examine its actual application effect for different people. We recruited three male volunteers aged 22–27 to complete this experiment. Among them, two subjects (marked as Subject 1 and Subject 2) who have taken part in the training data set collection for torque estimation, are selected to join the experimental group, and one (marked as Subject 3), who did not participate in that process, is assigned to the control group. It is worth noting that volunteers should not have done any high-intensity exercise 24 hours before the tests, to avoid affecting the physiological state of the muscles. During the experiment, they are told to exert an external force that changes approximately in accordance with the sine law on the measurement platform. All subjects knew and agreed with relevant experimental procedures in advance. Research related to this article was approved by the Laboratory Academic Committee of the State Key Laboratory of Robotics and System, Harbin Institute of Technology.

The embedded system, mounted on an upper extremity exosuit, calculates the estimated torque in real time, and sends them to a PC after being processed. The sensor on the measurement platform obtains the force data at the end of the arm, which is converted into the actual torque value in the PC. As it only aims to evaluate the reliability of torque estimation, we have shielded the subtask of the motor servo control in the program, so as to avoid the influence of man-machine coupling.

Figure 9 demonstrates the elbow joint torque estimation results of the upper extremity exosuit on three subjects. In the experimental group, it is obvious that the estimated torque looks very close to the actual value in terms of magnitude and variation trend. Under this condition, the model performance behaves in a relatively stable manner, and the identification result remains rather accurate; however, in the control group, the estimated torque cannot effectively follow the change of the actual value.

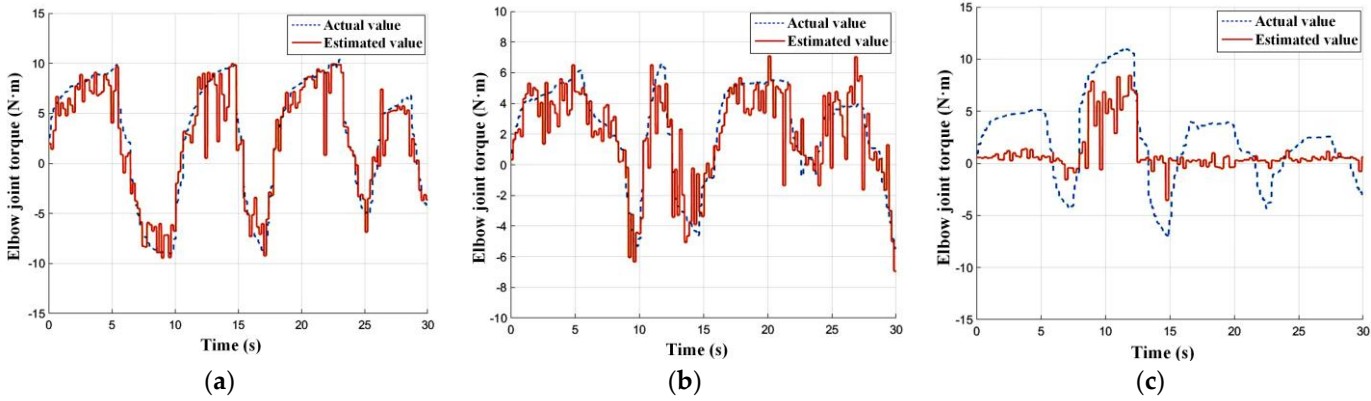

**Figure 9.** Elbow joint torque estimation results: (**a**) experimental result of Subject 1; (**b**) experimental result of Subject 2; (**c**) experimental result of Subject 3.

We use the *RMSE* and $R^2$ introduced above to quantitatively describe the identification effect for different subjects. It can be seen from Table 1 that the *RMSE* of the experimental group is lower than that of the control group, indicating that the error between the actual and estimated value is smaller for the joint torque of Subject 1 and Subject 2. Moreover, the $R^2$ in the experimental group comes up to 100%, which, when closely compared with the control group, means that the trained RFR model can perform better when utilizing the biological signals of Subject 1 and Subject 2.

**Table 1.** Evaluation results of torque estimation on three subjects.

| Groups | Participants | *RMSE* | $R^2$ |
|---|---|---|---|
| Experimental group | Subject 1 | 1.9812 | 0.9532 |
| | Subject 2 | 1.7008 | 0.8620 |
| Control group | Subject 3 | 3.4261 | 0.6824 |

From the above qualitative and quantitative description, the following two conclusions can be obtained.

- If we have collected a person's MMG signal for training the RFR model offline, the reliability of the online torque estimation will remain at a pretty high level when they wear the exosuit that uses the trained model;
- utilizing a trained model to estimate the joint torque of unknown subjects online may significantly weaken the effectiveness of identification.

The reason may be that muscle activation varies among different people when they output the same joint torque, or different thicknesses of adipose layers more or less influences MMG propagation; therefore, when using the exosuit for power assistance, it is necessary to independently train a matching torque estimation model for the wearer based on his/her biological information.

### 5.2. Efficiency Evaluation Experiment for Power Assistance

In order to verify the actual power-assisted effect of this method, we selected a healthy subject, and collected his MMG signals at the brachioradialis, deltoid, and ectopectoralis to customize a set of RFR models for him. After transplanting these trained models to the embedded system, and enabling all the subtasks of control program, the subject wears the exosuit to perform elbow static flexion/extension, shoulder static flexion/extension, and shoulder static adduction/abduction on the measurement platform, and tries to complete three evaluation experiments. Other conditions and requirements are basically the same as the above experiment. An emergency stop switch needs to be held by the right hand all the way through the experiment, to ensure that the experiment can be stopped in time if an accident occurs.

Figure 10 shows the performance evaluation experiments for joint movement assistance. We take three torque values, which are estimated by the RFR model, calculated by the tension sensor on the cable, and converted by the six-dimension force sensor on the measurement platform as human-exerted, exosuit-generated, and the total output, respectively.

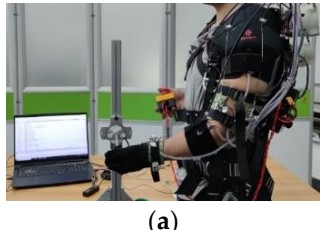 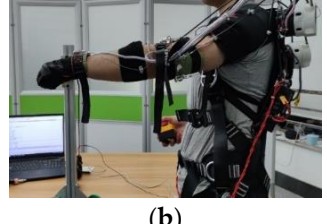 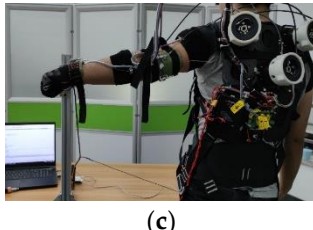

（**a**）　　　　　　　　　　　（**b**）　　　　　　　　　　　（**c**）

**Figure 10.** Actual power-assisted experiments: (**a**) experiment for elbow static flexion/extension; (**b**) experiment for shoulder static flexion/extension; (**c**) experiment for shoulder static adduction/abduction.

Figure 11 describes the changing situation of different torques in the typical time period of each motion mode. Obviously, it can be seen that the upper extremity exosuit can produce additional assistance in the three degrees of freedom of the shoulder and elbow joints, although its actual output is smaller than the torque estimated by the physiological signal. This error can be attributed to the loss of power transmission caused by friction between cable and sheath, or the calculation model deviation induced by suit deformation.

Moreover, the total output far exceeds the human effort, indicating that this exosuit can significantly enhance joint strength.

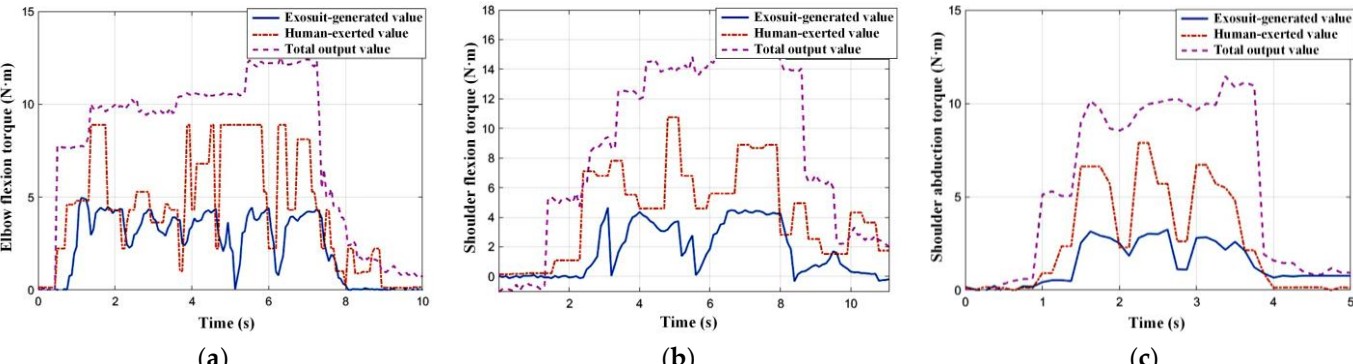

**Figure 11.** Variations of different torques in each motion mode: (**a**) torque curves for elbow static flexion; (**b**) torque curves for shoulder static flexion; (**c**) torque curves for shoulder static abduction.

We introduce the sum of absolute values (*ASUM*) to quantitatively reflect the average level of the three torques in different motion modes, and the corresponding results are shown in Figure 12.

$$ASUM = \frac{1}{n} \sum_{i=0}^{n} |T_i| \tag{13}$$

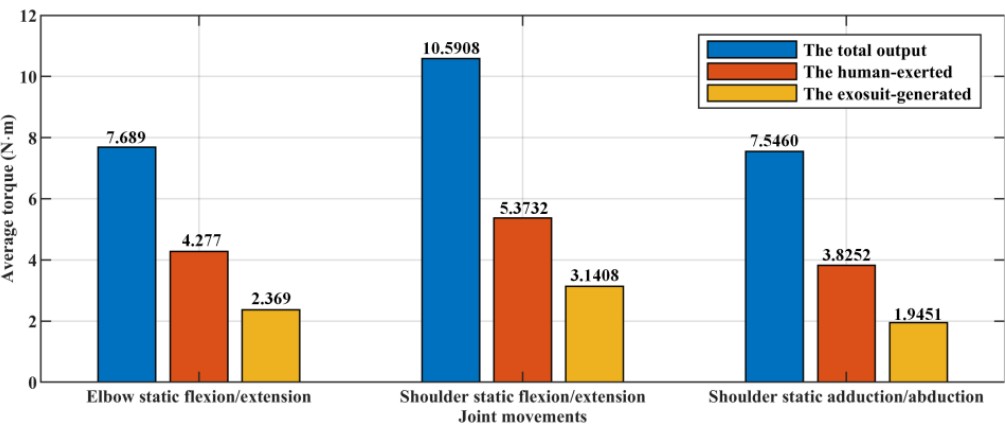

**Figure 12.** *ASUM* of different torques under each joint movement.

It is obvious that the sum of torque generated by the exosuit, and that exerted by a human, is not equal to the actual total output. The combined effect of factors such as identification error, transmission error, and calculation error, may lead to this gap between the expected and the actual. We expect to describe the power-assisted efficiency (*P*) through analyzing the ratio of $ASUM_{exosuit}$ to $ASUM_{total}$.

$$P = \frac{ASUM_{exosuit}}{ASUM_{total}} \tag{14}$$

The calculation results show that when the upper extremity exosuit independently assists elbow static flexion/extension, shoulder static flexion/extension, and shoulder static adduction/abduction, the corresponding power-assisted efficiencies come up to 30.81%, 29.66%, and 25.78%, respectively. These data mean that when a person is equipped with this wearable robot, the output torque for each joint of the upper limb can be roughly reduced by a quarter to a third.

## 6. Conclusions and Future Work

In this article, we propose a MMG-based joint torque estimation algorithm which realizes the decoding from biological signals to limb strength, and transplant it into the control system of an exosuit to calculate the motor commands for assisting the multi-joint motion of the upper limb. Two sets of experiments are carried out to test the reliability of torque estimation and the efficiency of power assistance.

The data collection and signal processing methods used in this paper effectively establish the data sets which reflect human body information. The specially designed measurement platform can obtain the MMG of muscles and corresponding joint torque in a relatively accurate and convenient way. The HHT successfully eliminates the interference components in the original MMG signal, and lays a solid foundation for future extraction.

A technical approach to estimate joint torque from the MMG signal is built through training the RFR model offline. The reliability analysis experiment shows that this method can enable the exosuit to accurately identify the wearer's current joint torque, but the model parameters need to be specially trained for everyone.

A torque estimation-based control strategy is successfully applied to the motion control of the upper extremity exosuit. The efficiency evaluation experiment indicates that the exosuit using this algorithm can significantly enhance the limb strength of wearers.

Based on the actual execution of the research, we believe that the current work has the following limitations. First of all, the nonlinear disturbance caused by transmission friction and motion hysteresis significantly reduces the control performance and power-assisted efficiency of the upper extremity exosuit. In addition, the MMG-based torque estimation algorithm has limitations in its application. The model parameters may be trained separately for each person, and even each muscle.

Therefore, future work and research directions should aim to break through the above limitations. First, an error compensation algorithm for this cable-driven system should be introduced into the control logic, in an attempt to offset interferences caused by nonlinear characteristics. Second, more general intention recognition algorithms need to be studied further, which can meet the usage requirements of every wearer without additional training preparation.

**Author Contributions:** Methodology, data analysis, writing, Y.S.; conceptualization, project administration, W.D.; experimental verification, data preprocessing, W.L.; resources, supervision, L.H.; schematic analysis, resources, X.W.; structure design, resources, P.L.; formal analysis, software, Y.G. All authors have read and agreed to the published version of the manuscript.

**Funding:** This work was funded by the pre-research project in the manned spaceflight field of China (Project Number 020202).

**Informed Consent Statement:** Informed consent was obtained from all subjects involved in the study.

**Acknowledgments:** The authors thank all those who have provided suggestions and assistance for this research and paper.

**Conflicts of Interest:** The authors declare no conflict of interest.

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
