# Peer review of "Human Joint Torque Estimation Based on Mechanomyography for Upper Extremity Exosuit"

_electronics, doi:10.3390/electronics11091335_

Round 1

Reviewer 1 Report

The paper presents a human joint torque estimation method based on mechanomyography. The authors are strongly encouraged to take action regarding the following feedback:

  1. The contribution of this paper/study to the field is unclear. What is innovative? What problem did you solve? How much performance did you improve?
  2. The technical approach is questionable: you need the human subjects to hold the 6-DOF F/T sensor in order to collect data - this has a big limitation as normally people do not hold their fists while executing tasks. Also, you mentioned that the subjects "try to perform specific actions", this is also different from what human really does in elbow flexion and extension. The muscle volume/shape would change during elbow flexion/extension, but you ignore it, could you provide more detailed reasoning behind this simplification?
  3. You recruited 3 human subjects for the experiment but did not provide the corresponding IRB approval.
  4. It does not make too much sense to me that you are able to obtain 150000 sets of sample data from 3 subjects. How do you define a "set" of data? How much time did you spend on collecting data?

Reviewer 2 Report

This paper focused on the issue of human intention recognition to control exoskeleton robots and provide corresponding assistance. This paper presents a mechanomyography (MMG) based method to estimate the current joint torque and provide rehabilitation assistance using the upper extremity exosuit. Hilbert-Huang Transform (HHT) is introduced to process this biological signal and eliminate the mixed noise and motion artifacts. A torque estimation-based control strategy is developed for the intention understanding and motion servo of a customized upper extremity exosuit. This research work is interesting for the soft robot control research society. However, this paper has several limitations and the standard is not enough, and address the following items would result in a good paper,

  1. The literature review is not thorough about the application and the contributions. To highlight the contributions, it suggests reorganizing the section of the related work. It is recommended to read the following works and consider to discuss their application scenarios in the introduction and discussion, Improved recurrent neural network-based manipulator control with remote center of motion constraints: Experimental results; Multi-Sensor Guided Hand Gesture Recognition for a Teleoperated Robot Using a Recurrent Neural Network
  2. The contribution of this paper is not clear. It suggests revising the contributions section and making these points clear and strong.
  3. The quality of Figure 1 should be improved and readable for the readers.
  4. Maybe it is better to discuss the possibility to improve the scope using deep learning to learn and optimize in the introduction, for example, A Multimodal Wearable System for Continuous and Real-time Breathing Pattern Monitoring During Daily Activity, DCNN based human activity recognition framework with depth vision guiding
  5. It is recommended to present in the first section so that it can highlight the specific scope of this article. The meaning of the assessment experiment should be highlighted.
  6. There should be a further discussion about the limitation of the current works, in particular, what could be the challenge for its related applications. To let readers better understand future work, please give specific research directions.

Reviewer 3 Report

Abstract: Well written but would significantly benefit from clearly stating your, key findings, contributions and the novelty of this research. Currently the abstract reads like a condensed introduction rather than an overview.

Introduction: 

Some grammatical errors here that make reading difficult, I would consider more concise sentences.

An explanation of EEG would be beneficial for a broader audience, another area to consider is eye tracking and GSR (https://www.researchgate.net/publication/299467637_A_method_to_control_bionic_arm_using_Galvanic_skin_response), I would include these as you do not have a related works section. It is worth discussing these different approaches in the introduction to justify your methodology.

Good discussion of the limitations of EEG in practice in the introduction, i.e connector conductivity during movement, etc. Dose MMG pose similar issues?  you state lots of reasons why this method is suitable I think it may be pertinent to state any potential limitations that may cause adverse sensor feedback ect.

A very good methodology section, really considered, I think your work on the suit is exceptional. I think the sections on calibration and testing is of significant interest, good and appropriate use of RFR. 

Why do you only use a sample of 3 individuals? this is very very low, please justify this in the paper. i.e the amount and complexity of data you are collecting would not be feasible for a larger sample. 

Good discussion of system limitations.

The paper would benefit from a future works section.

Round 2

Reviewer 1 Report

R1 #2:

No, your response does not make too much sense to me. You already said, "it is a common idea to measure ... during dynamic joint movements ..." and you "do not have such a basis" does not lead to the conclusion that your simplified method is valid. The logic behind it is very questionable. If you do not have the equipment, then please purchase one.

R1 #3:

"Laboratory Academic Committee of the State Key Laboratory of Robotics and System, Harbin Institute of Technology" does not sound like an IRB, do you know what an IRB should be? Please also include your IRB approval number.

R1 #4:

Your response does not explain how 150000 sets of sample data could be collected from 3 subjects. Please provide more detailed information. Please also provide the data you collected.

Reviewer 2 Report

The authors have addressed all of my concerns. The current version can be accepted now. No more revision is required from my side. 

Author Response

Thanks for your valuable comments on this article. Thanks for acknowledging this research work and related revisions of the manuscript. We will continue to carry out related researches in the field of wearable robots.